# CONFIDENCE IS ALL YOU NEED: FEW-SHOT RL FINE-TUNING OF LANGUAGE MODELS

## ABSTRACT

Large Language Models (LLMs) have demonstrated strong performance on reasoning tasks, but post-training optimization remains essential for aligning their behavior with specific task objectives. Existing reinforcement learning (RL) approaches often rely on costly human annotations or external reward models, limiting their scalability in real-world applications. To address this, we propose Reinforcement Learning via Self-Confidence (RLSC)—a method that uses the model's own confidence in its outputs as the reward signal, without requiring human labels, preference models, or manually crafted reward functions. RLSC is also highly sample-efficient: it only needs 1 to 8 samples per problem, and typically converges within 15 to 30 training steps. Under the Pass@1 evaluation metric, Qwen-Math-7B achieves significant performance improvements across several mathematical benchmarks: AIME2024 +6.7%, AMC23 +33.1%, Math500 +32.3%, Minerva +29.8%.On average, RLSC delivers a 23.68% improvement across these benchmarks. Notably, the effectiveness of RLSC is not limited to the Qwen series; it also leads to substantial performance gains on other mainstream models, including Olmo-7B, DeepSeek-R1-Distill-Qwen-1.5B, DeepSeek-R1-Distill-Llama-8B, Gemma-4B, and LLaMA-8B, etc. In summary, RLSC offers a simple, efficient, and scalable post-training method for pretrained language models, enabling significant performance gains with few training steps.

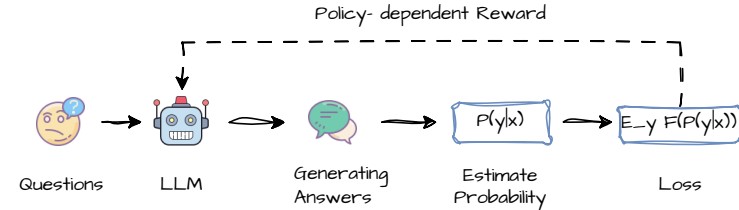

(a) Overview of the Reinforcement Learning via Self Confidence (RLSC) approach.

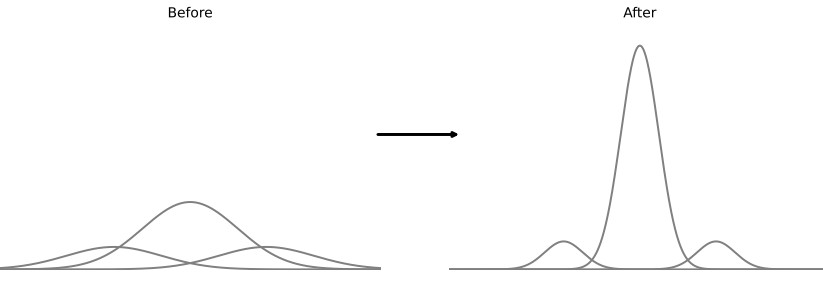

(b) Response probability distribution.

Figure 1: Combined visualization: (a) RL via Self Confidence workflow schema; (b) Probability distribution before and after training.

# 1 INTRODUCTION

Large language models Achiam et al. (2023) Bai et al. (2023) Yang et al. (2024b) Liu et al. (2024) Guo et al. (2025) Guo et al. (2024) Grattafiori et al. (2024) have demonstrated remarkable capabilities across various domains. However, to further align model behavior with specific task objectives, post-training optimization remains indispensable. Compared to supervised fine-tuning, reinforcement learning (RL) offers stronger exploration capabilities and is thus widely employed to enhance the reasoning performance of large models. Currently, reinforcement learning algorithms such as DPO Rafailov et al. (2023) , PPO Schulman et al. (2017), RLHF Ouyang et al. (2022), and GRPO Guo et al. (2025) are extensively applied in the post-training of large language models. Among these, Reinforcement Learning with Verifiable Rewards (RLVR) Zeng et al. (2025) Liu et al. (2025) has become the mainstream approach for defining reward signals in RL training for large language models, significantly simplifying the complexity of reward design and improving training controllability and efficiency.

Although large language models have achieved remarkable progress in mathematical reasoning tasks, reinforcement learning (RL)-based fine-tuning methods often rely on costly human-labeled datasets or carefully designed reward functions. Performing RL training without labels could significantly reduce the cost of post-training optimization. Test-Time Reinforcement Learning (TTRL) Zuo et al. (2025) addresses this by generating up to 64 candidate answers per question and using majority voting to create pseudo-labels for training. However, this approach incurs substantial computational overhead, which limitsmits its practicality for large-scale models.

To address the above challenges, we propose a new paradigm: *Reinforcement Learning via Self-Confidence* (**RLSC**). Unlike methods that rely on external supervision—such as verifiable rewards or pseudo-label generation in TTRL, RLSC uses the model's own confidence in its outputs as a reward signal, eliminating the need for any external supervision or manually constructed labels. By combining confidence analysis, the internal knowledge embedded within pre-trained language models can be effectively leveraged, enabling improved performance on downstream tasks—even without any external feedback.

We validate the proposed RLSC method on multiple models, including Qwen2.5-Math Yang et al. (2024b), Gemma Team et al. (2024), LLaMA Grattafiori et al. (2024), the distilled version of DeepSeek R1 Guo et al. (2025), and Olmo Groeneveld et al. (2024). Training was conducted on datasets such as GSM8K Cobbe et al. (2021), AIME24 Li et al. (2024b), Math500 Hendrycks et al. (2021c), Minerva Math Hendrycks et al. (2021b), and AMC23 Li et al. (2024a), using only *15–30 training steps*. With a sampling temperature of 1, the model generated just 1 to 8 samples per problem and still achieved rapid performance gains.

All evaluations were performed under non-zero temperature sampling settings. RLSC showed particularly strong results on the Qwen2.5-Math-7B model: under the Pass@1 Guo et al. (2025) metric, it achieved a 6.67% improvement on AIME2024, 33.1% on AMC23, 32.3% on Math500, and 29.8% on Minerva Math. Across multiple math benchmarks, the average improvement reached 23.68%.

These results demonstrate that, when combined with the RLSC framework, a strong pre-trained language model can achieve significant gains in confidence and generalization—without the need for auxiliary datasets, human feedback, or handcrafted reward functions, and with only a few steps of training.

**Our main contributions are:**

1. **A novel training paradigm:** We propose RLSC for the first time—an externally-supervision-free reinforcement learning method that uses the model's own output confidence as a reward signal for self-supervised training.

2. **No reliance on labels or external rewards:** Unlike existing methods such as RLHF, DPO, PPO, GRPO, or TTRL, RLSC completely eliminates the need for costly human labels, reward models, or complex reward shaping, significantly reducing the cost of post-training.

3. **High training efficiency:** RLSC achieves convergence with only a few training steps (15–30) and a small number of samples (just 1 to 8 per problem), and can be fine-tuned in 30 minutes on a single 8×A100 GPU cluster using major LLMs such as Qwen2.5-Math and Gemma, etc.

4. **Outstanding generalization:** RLSC demonstrates significant performance improvements on several mathematical benchmarks (AIME24, AMC23, Math500, Minerva, etc.), with an average gain of 23.68%, maintaining strong performance on Pass@1 settings.

## 2 METHOD

### 2.1 FROM TTRL TO MODE SHARPENING

Test-Time Reinforcement Learning (TTRL) Zuo et al. (2025) improves large language models (LLMs) by generating many outputs per input (typically 64) and applying majority voting to select the most frequent completion. This pseudo-label is then used to fine-tune the model. Although this method has proven effective by experiments, it incurs substantial computational overhead during training.

Inspired by the idea of majority voting, we asked the following question:

*what is the key underlying principle behind this voting process?*

Intuitively, majority voting selects the mode of the output distribution. Optimizing for agreement between sampled completions sharpens the distribution: it increases the probability mass concentrated on the most likely answer. This also increases the chance of generating the same answer while taking two independent samples.

Let $p_\theta(y \mid x)$ denote the model probability of generating response $y$ given input $x$, parameterized by $\theta$. The probability that two independent samples from this distribution are identical is:

$$F(p_\theta) = \mathbb{E}_{y_1, y_2 \sim p_\theta(y|x)}[\mathbb{I}(y_1 = y_2)] = \sum_y p_\theta(y \mid x)^2 \qquad (1)$$

This expression is maximized when the distribution collapses to a delta function centered on a single most probable response - i.e., when the model is confident.

Therefore, we propose to directly maximize the following self-confidence objective:

$$F(p_\theta) = \mathbb{E}_{y \sim p_\theta(y|x)}[p_\theta(y \mid x)] \qquad (2)$$

This objective acts as a continuation of the TTRL method. We suggest that it has similar effect on probability distribution of model's answers, while eliminating the need of generating large samples that are needed for accurate majority voting. The key advantage is that it can work with any sample size, even with a single output, and allows straightforward calculation based on model's output with no need for extra computation.

### 2.2 SELF-CONFIDENCE LOSS AND GRADIENT

To optimize the self-confidence objective 2, we compute its gradient with respect to the model's parameters $\theta$. Applying the log-trick, we obtain:

$$\begin{aligned}
\nabla_\theta F(p_\theta) &= \sum_y \nabla_\theta p_\theta(y \mid x) \, p_\theta(y \mid x) \\
&= \mathbb{E}_{y \sim p_\theta}\left[\nabla_\theta p_\theta(y \mid x)\right] \\
&= \mathbb{E}_{y \sim p_\theta}\left[p_{old}(y \mid x) \, \nabla_\theta \log p_\theta(y \mid x)\right]
\end{aligned} \qquad (3)$$

Here, $p_{old}(y \mid x)$ represents the probability of completion as it was at the time of model sampling. As the rollouts are samples from the distribution by $y \sim p_\theta$, we can define the loss function as:

$$\mathcal{L}_1 = -\sum_y p_{old}(y \mid x) \log p_\theta(y \mid x) \qquad (4)$$

This loss promotes higher log-probabilities for responses to which the old model assigned higher confidence. It does not require an external reward model or labels, relying solely on the model's own confidence as feedback. This enables the use of cheaper, unlabeled data.

We also generalize this to a broader class of differentiable functions $\mathcal{L}(p_{\text{old}}, p_\theta)$. An effective variant smooths the weighting by adding a constant $\alpha > 0$:

$$\mathcal{L}_2 = - \sum_y \big(p_{old}(y \mid x) + \alpha\big) \log p_\theta(y \mid x) \tag{5}$$

We empirically find that even small values of $\alpha$ (for example, $0.1$) improve both convergence and generalization.

Table 1: Loss functions and corresponding optimized functionals

| Name | Loss function | Functional |
|------|---------------|------------|
| RLSC loss | $p_{\text{old}} \log p$ | $\mathbb{E}_{p_\theta}[p_\theta]$ |
| Shannon Entropy | $(1 + \log p_{\text{old}}) \log p$ | $\mathbb{E}_{p_\theta}[\log p_\theta]$ |
| Advantage Score | $A(y) \log p$ | $\mathbb{E}_{p_\theta}[A(y)]$ |

Some alternative estimation methods and their corresponding loss functions are also proposed, as shown in Table 1.

These statements capture the core idea and motivation behind the RLSC method, with its algorithm presented in pseudocode in Algorithm 1.

---

**Algorithm 1** The pseudo-code of the RLSC for LLM

---

```
# model.generate(prompt): generates multiple completions
# model.forward(input): returns token logits

for question in dataset:

    # generate completions
    completions = model.generate(question, temperature, num_samples)

    # get gradable probabilities
    logits = model.forward(question.repeat() + completions)[question.
        length:-1]
    all_log_probs = log_softmax(logits / temperature)
    log_p = all_log_probs.gather(token_ids).sum

    # compute loss
    loss = - (exp(log_p).detach() + alpha) * log_p

    loss.backward()
    optimizer.step()
```

---

## 3 EXPERIMENTS

### 3.1 TRAINING SETUP

We leverage the self-confidence signal as a training supervision mechanism to fine-tune a range of models, including Qwen Yang et al. (2024b), LLaMA Grattafiori et al. (2024), Gemma Team et al. (2024), Olmo Groeneveld et al. (2024), and distilled versions of Deepseek-R1 Guo et al. (2025), among others. The training procedure is as follows:

1. For each question, call generate(temperature = 0.6, num of samples = n) to produce n candidate responses (with n = 1 to 8);

2. For each (prompt + response) pair, tokenize the sequence and compute token-level log-probabilities. Then, resample the tokens with a temperature of 1 to ensure the original distribution remains unchanged;

3. Apply an assistant mask to keep only the response tokens, and sum the log-probabilities across the masked region to obtain the sequence-level response likelihood;

4. Compute the loss based on the sequence-level log-likelihood and update model parameters via backpropagation using the Adam optimizer.

**Training Dataset.** We performed reinforcement learning-based training on the GSM8K Cobbe et al. (2021), AIME24 Li et al. (2024b), AMC23 Li et al. (2024a) and MATH500 Hendrycks et al. (2021c) datasets.

## 3.2 EVALUATION AND RESULTS

**Benchmarks.** We evaluated our method on several challenging benchmark datasets, including mathematical reasoning tasks such as AIME24 Li et al. (2024b), MATH500 Hendrycks et al. (2021c), AMC23 Li et al. (2024a), MMLU STEM Hendrycks et al. (2021a), Minerva Math Hendrycks et al. (2021b), Olympiadbench He et al. (2024).

**Evaluation Metrics.** For model evaluation, we adopt the Pass@1 metric as proposed in DeepSeek-R1 Guo et al. (2025), which measures the average accuracy under high-temperature and multi-sample generation settings, as defined in Equation 6.

**Evaluation Metrics.** For model evaluation, we adopt the **Pass@1** metric as proposed in DeepSeek-R1 Guo et al. (2025), which measures the average accuracy under high-temperature and multi-sample generation settings, as defined in Equation 6. All evaluations are conducted using a generation setting of 3072 new tokens, with **temperature** = 0.6 and **top-p** = 0.95.

$$\text{Pass@1} = \frac{1}{k} \sum_{i=1}^{k} p_i \tag{6}$$

To ensure a fair comparison, we evaluated both our trained model checkpoints and the baseline using the same evaluation template, with all experimental settings kept identical, detailed evaluation procedures can be found in Appendix B.2.

Table 2: Performance comparison across different benchmarks (Pass@1). All compared models have the same size (7B). For each benchmark, the **best** result is highlighted in bold, the second best is underlined, and the *third best* is italicized, * indicates the best-performing result among models trained without labeled data. Eurus-2-7B-SFT is developed based on Qwen2.5-Math.

| Model | Base | Reward | AIME24 | MATH500 | AMC23 | MMLU | Minerva | Olympiad | Avg |
|---|---|---|---|---|---|---|---|---|---|
| *With using extra labeled data.* | | | | | | | | | |
| Qwen2.5-Math-7B-Instruct Yang et al. (2024b) | * | SFT+RL | 8.8 | **81.6** | *58.8* | **70.3** | 38.8 | 37.6 | 49.3 |
| Open-Reasoner-Zero Hu et al. (2025) | Qwen2.5 | Rule-Based | 13.3 | 79.6 | 51.3 | 68.1 | 38.2 | **41.2** | 48.6 |
| SimpleRL-Zoo Zeng et al. (2025) | Qwen2.5-Math | RLVR | 28.3 | 74.9 | 59.4 | 56.1 | 23.0 | 34.2 | 46.0 |
| Oat-Zero Liu et al. (2025) | Qwen2.5-Math | RLVR | **31.7** | 79.1 | **68.8** | 40.6 | 40.4 | 39.9 | *50.1* |
| Qwen2.5-Math-7B-GRPO* | Qwen2.5-Math | RLVR | *23.3* | 78.3 | 67.5 | 56.9 | **40.9** | 39.3 | 51.0 |
| *Without using any extra labeled data.* | | | | | | | | | |
| Qwen2.5-Math-7B (Base Line) Yang et al. (2024b) | * | * | 10.8 | 46.7 | 34.4 | 47.6 | 10.4 | 14.7 | 27.4 |
| Spurious Reward Shao et al. (2025) | Qwen2.5-Math | Spurious Reward | 10.8 | 50.9 | 41.9 | 48.4 | 9.7 | 17.8 | 29.9 |
| TTRL Zuo et al. (2025) | Qwen2.5-Math | Majority Voting | 18.3* | 74.4 | 58.8 | 54.8 | 36.9 | 37.0 | 46.7 |
| AZR Zhao et al. (2025) | Qwen2.5 | Self Reward | 11.7 | 64.1 | 39.4 | 62.9 | 18.3 | 29.5 | 37.6 |
| PRIME-Zero Cui et al. (2025) | Qwen2.5 | Implicit Reward | 16.7 | 55.1 | 45.0 | 48.9 | 10.5 | 19.5 | 32.6 |
| PRIME Cui et al. (2025) | Eurus-2-7B-SFT | Implicit Reward | 13.3 | 72.6 | 61.3 | 41.5 | 39.5 | 34.4 | 43.8 |
| **RLSC (Ours)** | Qwen2.5-Math | Self-Confidence | 17.5 | 79.0* | 67.5* | 63.1* | 40.2* | 39.4* | **51.1*** |

We evaluated the above models using the official Qwen Math evaluation scripts Yang et al. (2024a). Qwen2.5-Math-7B-GRPO* is our model, trained for 1 epoch on the Deepscaler Luo et al. (2025) dataset with the GRPO and RLVR mechanisms. Except for the Spurious-Reward Shao et al. (2025) and TTRL Zuo et al. (2025) models, which we re-implemented based on the scripts released with the Spurious-Reward paper (see training details in Appendix B.1), all other models were evaluated

using trained weights from HuggingFace. The evaluation parameters were kept consistent across all models.

As shown in Table 2, our method achieves the highest average performance across five mathematical benchmarks and one benchmark encompassing Science, Technology, Engineering, and Mathematics (STEM), including some models trained with labeled data. Notably, among models trained without labels, our approach outperforms existing methods on the MATH500, AMC23, MMLU Stem, Minerva, and Olympiad benchmarks.

Regarding training efficiency, most other models typically require at least five hours to train, with TTRL being particularly time-consuming—training on an 8×A100 setup took 15 hours due to its Majority Voting procedure (64 responses per prompt). In contrast, our method requires only 15–30 training steps, completing in approximately 30 minutes on the same 8×A100 GPU setup.

Table 3: Pass@1 performance comparison across different models. Evaluation was conducted with non-zero temperature sampling, averaging 4 responses per question.

| Model | AIME24 | MATH500 | AMC23 | MMLU | Minerva | Olympiad | Avg |
|---|---|---|---|---|---|---|---|
| *LLaMA-8B* | | | | | | | |
| LLaMA-8B Grattafiori et al. (2024) | 0.0 | 12.0 | 1.9 | 45.6 | **6.4** | 3.1 | 11.5 |
| Ours | **0.0** | **12.3** | **7.5** | **46.9** | 6.0 | **3.5** | **12.7** |
| Δ | **+0.0** | **+0.3** | **+5.6** | **+1.3** | **-0.4** | **+0.4** | **+1.2** |
| *LLaMA-8B-It* | | | | | | | |
| LLaMA-8B-It Grattafiori et al. (2024) | 3.3 | 49.5 | 19.4 | 49.4 | 19.1 | **17.6** | 26.38 |
| Ours | **8.3** | **49.6** | **23.1** | **50.4** | **24.0** | 17.1 | **28.75** |
| Δ | **+5.0** | **+0.1** | **+3.7** | **+1.0** | **+4.9** | **-0.5** | **+2.37** |
| *OLMo-2-7B* | | | | | | | |
| OLMo-2-7B Groeneveld et al. (2024) | 3.3 | 9.7 | 5.0 | 40.9 | 2.2 | 2.3 | 10.57 |
| Ours | **3.3** | **19.5** | **17.5** | **45.9** | **4.2** | **4.6** | **15.83** |
| Δ | **+0.0** | **+9.8** | **+12.5** | **+5.0** | **+2.0** | **+2.3** | **+5.26** |
| *Gemma-4B-Pt* | | | | | | | |
| Gemma-4B-pt Team et al. (2024) | 0.0 | 7.4 | 3.8 | 36.4 | 2.8 | 2.2 | 8.77 |
| Ours | **0.8** | **14.1** | **8.6** | **40.0** | **5.1** | **3.7** | **12.05** |
| Δ | **+0.8** | **+6.7** | **+4.8** | **+3.6** | **+2.3** | **+1.5** | **+3.28** |
| *Qwen-Math-1.5B* | | | | | | | |
| Qwen-Math-1.5B Yang et al. (2024b) | 5.0 | 34.8 | 30.0 | 33.3 | 8.7 | 19.9 | 21.95 |
| Ours | **6.7** | **66.5** | **43.1** | **48.1** | **26.2** | **28.7** | **36.55** |
| Δ | **+1.7** | **+31.7** | **+13.1** | **+14.8** | **+17.5** | **+8.8** | **+14.6** |
| *Qwen-Math-7B* | | | | | | | |
| Qwen-Math-7B Yang et al. (2024b) | 10.8 | 46.7 | 34.4 | 47.6 | 10.4 | 14.7 | 27.44 |
| Ours | **17.5** | **79.0** | **67.5** | **63.1** | **40.2** | **39.4** | **51.12** |
| Δ | **+6.7** | **+32.3** | **+33.1** | **+15.5** | **+29.8** | **+24.7** | **+23.68** |

The results are shown in Table 3. In multiple benchmark tests, all models achieved significant improvements, with particularly outstanding performance in the Qwen2.5-Math-7B (an average improvement of 23.68%). Even when the initial model performed poorly, effective performance gains were still realized.

### 3.3 ANALYSIS OF EMERGENT BEHAVIOR

**Impact of Training Datasets.** We found that different training datasets lead to significant variations in model performance. Through analysis, we believe that simpler datasets are more effective in enhancing the balance between the model's confidence and accuracy. As shown in Figure 2, we used greedy decoding and report only one generated response per prompt.

**More Confident Responses.** We observed that fine-tuning based on RLSC enables the model to produce answers that are more concise and confident. Unlike traditional fine-tuning methods, our model learns to identify the correct answer early, thereby avoiding lengthy and redundant reasoning.

**Analysis of Accuracy Improvements Across Different Models.** Experimental results demonstrate that the proposed method consistently improves performance across multiple models, yet a notable discrepancy arises between Qwen and LLaMA. Specifically, Qwen exhibits relatively low policy entropy at the early stage of training, indicating higher decision determinism, which enables the

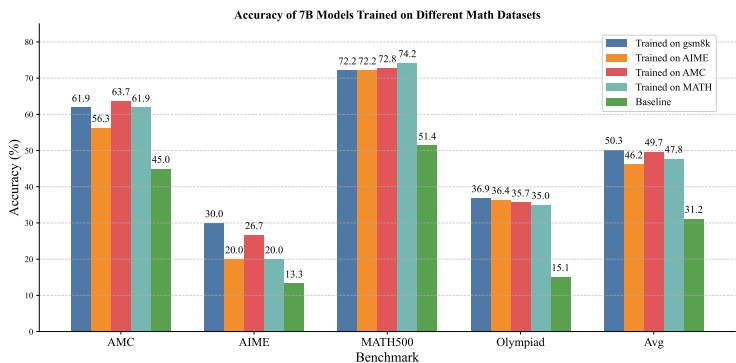

Figure 2: Comparison of the performance of models trained on various datasets, alongside the base-line.

sharpened distribution to enhance performance more rapidly. In contrast, LLaMA starts with significantly higher policy entropy, resulting in lower certainty. By further tracking the training process, we observe that ***LLaMA tends to improve its confidence by shortening the generated sequence length***, thereby gradually achieving convergence. This phenomenon suggests that the distinct policy entropy characteristics at the beginning of training may be a key factor underlying the differences in enhancement effectiveness across models.

**Visualization of Logits Distribution.** We visualized the logits distribution during the sampling process before and after training as shown in Figure 3. The results show that the variance of the distribution is relatively small before training, while it becomes significantly larger after training. Meanwhile, the red dashed line in the figure indicates the position of the maximum value. It can be clearly observed that the model becomes more confident in its predictions after training.

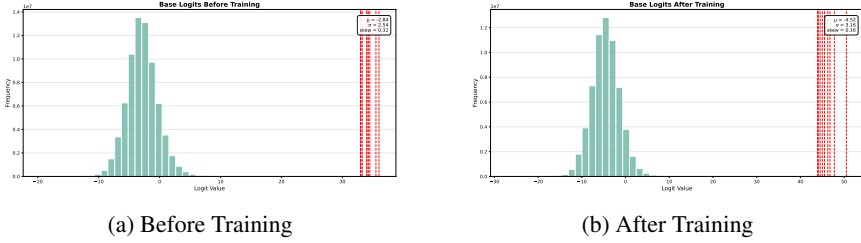

(a) Before Training            (b) After Training

Figure 3: Logits distribution before and after training. Red dashed lines indicate the top 10 maximum values.

**Training Efficiency: Model Average Accuracy Across All Steps.** Since our model is trained using an on-policy approach and gradients are computed based on entire sequences level rather than tokens level, it converges relatively quickly. Moreover, with 4 rollouts per prompt, the model achieves rapid convergence. As shown in Figure 4, Qwen2.5-Math-1.5B converges in 20 steps, while in the same configuration, other models are also able to converge in 15 to 30 steps.

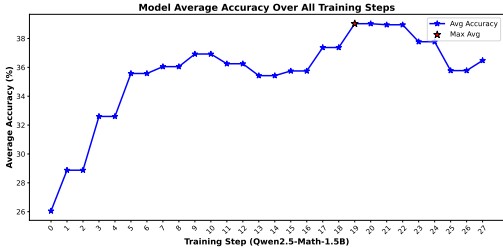

Figure 4: Average performance during training over 4 benchmarks.

**Confidence and Entropy.** As shown in Figure 5, the model's average confidence steadily increases during training, while the entropy decreases accordingly. This visualization indicates that the objective of sharpening the distribution has been successfully achieved.

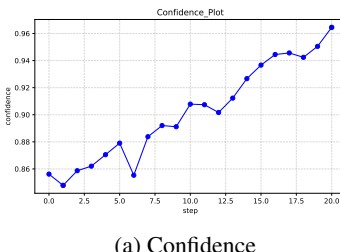

(a) Confidence

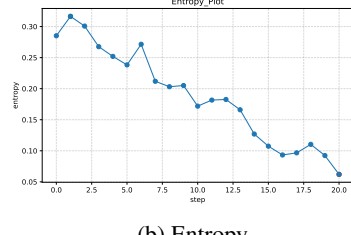

(b) Entropy

Figure 5: Variation of Average Confidence and Entropy Across Training Steps (Train on gsm8k).

**Cherry Pick.** Additionally, we present the outcomes of the two cases on AMC 2023 and AIME 2024, providing a clear, step-by-step exposition of the solution strategies.

---

### Case 1: Meeting Point of Two Bikers

**Problem.** Cities $A$ and $B$ are 45 miles apart. Alicia lives in $A$ and Beth lives in $B$. Alicia bikes towards $B$ at 18 miles per hour. Leaving at the same time, Beth bikes toward $A$ at 12 miles per hour. How many miles from City $A$ will they be when they meet? Mark your solution with \boxed.

---

**Solution.** To determine how many miles from City $A$ Alicia and Beth will be when they meet, follow these steps:

1. **Combined speed:** Alicia bikes at 18 mph, Beth at 12 mph.

$$\text{Combined speed} = 18 + 12 = 30 \text{ mph}$$

2. **Time to meet:** The distance between the cities is 45 miles.

$$t = \frac{\text{Distance}}{\text{Combined speed}} = \frac{45}{30} = 1.5 \text{ hours}$$

3. **Distance Alicia travels:**

$$\text{Distance} = \text{Speed} \times \text{Time} = 18 \times 1.5 = 27 \text{ miles}$$

**Answer:** They will meet 27 miles from City $A$.

$$\boxed{27}$$

---

### Case 2: Finding $xy$ from Logarithmic Equations

**Problem.** There exist real numbers $x$ and $y$, both greater than 1, such that

$$\log_x\left(y^x\right) = \log_y\left(x^{4y}\right) = 10.$$

Find $xy$. Mark your solution with \boxed.

---

**Solution.** Solve step by step.

1. Rewrite the logarithmic equations using logarithm properties:

$$\log_x(y^x) = x\log_x(y) = 10, \quad \log_y(x^{4y}) = 4y\log_y(x) = 10.$$

2. Let $\log_x(y) = a$ and $\log_y(x) = b$:

$$x \cdot a = 10 \implies x = \frac{10}{a}, \quad 4y \cdot b = 10 \implies y = \frac{5}{2b}.$$

3. Using the change-of-base formula:

$$a = \frac{\log(y)}{\log(x)}, \quad b = \frac{\log(x)}{\log(y)}.$$

4. Solving numerically gives

$$xy = 25.$$

**Answer:**

$$\boxed{25}$$

## 4 RELATED WORK

**Reinforcement Learning in Reasoning Tasks.** In recent years, **Reinforcement Learning (RL)** has played a pivotal role in enhancing the reasoning capabilities of Large Language Models (LLMs). Models such as DeepSeek-R1 Guo et al. (2025), ChatGPT o1 Achiam et al. (2023), QwQ Team (2025), and Qwen have demonstrated impressive reasoning skills by decomposing complex problems into intermediate steps and engaging in deep deliberation prior to producing final responses capabilities often acquired and refined through reward-driven learning mechanisms.

A widely adopted paradigm is Reinforcement Learning from Human Feedback (RLHF) Ouyang et al. (2022), which aligns model behavior with human preferences using human annotations or learned preference models as reward signals. RLHF typically employs Proximal Policy Optimization (PPO) Schulman et al. (2017) for policy updates. However, RLHF is heavily reliant on costly human annotations, which limits scalability.

To alleviate this dependency, Reinforcement Learning with Verifiable Rewards (RLVR) Luong et al. (2024) Lambert et al. (2024) Zeng et al. (2025) Liu et al. (2025) adopts a paradigm where rewards are computed directly from question–answer pairs $(x, y^*)$, by comparing model outputs with reference answers. By leveraging such a scheme, RLVR greatly reduces the resource requirements of training, though it still depends on human-labeled questions and answers, which constrains scalability.

**Test-Time Training.** More recently, **Test-Time Training (TTT)** Zuo et al. (2025) has emerged as a promising direction for further optimizing model behavior during inference. Notable examples include *SelfPlay Critic (SPC)* Chen et al. (2025) and *Absolute Zero Reasoner (AZR)* Zhao et al. (2025), which employ adversarial dual-model frameworks inspired by game-theoretic learning. In these approaches, one model acts as a "sneaky generator" crafting challenging or misleading reasoning steps, while the other serves as a "critic" that learns to detect errors. These methods eliminate the need for human supervision but depend on external tools (e.g., Python executors or code verifiers) to supply feedback signals.

Another prominent TTT approach is TTRL Zuo et al. (2025). It constructs pseudo-labels by sampling multiple candidate responses for each question and applying a majority-vote mechanism. The resulting consensus serves as a proxy label to compute rewards for model updates. Although TTRL avoids explicit human supervision, it requires a large number of samples (e.g., 64 per question), which leads to significant computational overhead.

**Summary and Motivation.** The reliance on annotations, external models, or hand-crafted rewards underscores the need for a simple and efficient label-free reinforcement signal that can be derived directly from the model itself.

## 5 CONCLUSION

We introduced **Reinforcement Learning via Self-Confidence (RLSC)**, a lightweight fine-tuning method that eliminates the need for labels, preference models, or handcrafted rewards. Unlike prior approaches such as TTRL, which rely on large-scale majority voting, RLSC formalizes the underlying principle mathematically.

Our key contribution is a derivation: we show that majority voting implicitly optimizes for agreement within the model's output distribution - and we transform that into a differentiable, self-supervised objective. This "mode sharpening" functional enables efficient reinforcement learning directly from the model's own confidence.

We find that for nearly all pretrained models, RLSC converges reliably within 4–8 rollouts per question and 15–30 training steps, while still delivering substantial accuracy improvements without external supervision.

This work demonstrates that high-quality post-training can emerge not from external labels, but from a model's internal signal - when that signal is derived with care. We believe RLSC provides both a practical tool and a conceptual bridge between majority-voting-based pseudo-labeling and principled self-supervision.

ETHICS STATEMENT

This work strictly adheres to the ICLR Code of Ethics. The research does not involve human subjects, nor does it pose potential harm to society, individuals, or the environment. There are no conflicts of interest, nor any issues related to discrimination, bias, or fairness. All experiments and analyses were conducted following academic norms and ethical guidelines.

REPRODICIBILITY STATEMENT

All content reported in this paper is accurate. Detailed information on the evaluation and training methods can be found in the Supplementary Materials to facilitate reproducibility. The Supplementary Materials include data processing steps, model, training hyperparameters, and evaluation metrics.

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

# A USE OF LARGE LANGUAGE MODELS (LLMS)

We used a large language model (LLM) to assist with grammar correction and language polishing. All scientific content, analysis, and results were generated by the authors, and the LLM was only used for improving readability and clarity.

# B CONFIGURATION

## B.1 TRAINING CONFIGURATION

All experiments were performed on a cluster equipped with 8×NVIDIA A100 GPUs, ensuring a consistent hardware setup across all training runs.

### B.1.1 RLSC

We train the RLSC model with the following settings:

- **Batch size:** 1
- **Generation per question:** 1–8 responses (Rollouts)
- **Top-p sampling:** 0.95
- **Temperature:** 0.95
- **Repetition penalty:** 1.05
- **Total training steps:** 30
- **Generated sequence length:** 512 or 768 or 1024 tokens
- **Gradient accumulation:** 1 or 2
- **Optimizer:** AdamW
- **Learning rate:** $5 \times 10^{-6}$
- **Reward:** RLSC

All hyperparameters were kept consistent across experiments, unless otherwise specified.

### B.1.2 QWEN2.5-MATH-7B-GRPO*

We train the Qwen2.5-Math-7B-GRPO* model with the following settings:

- **Training dataset:** DeepScaleR
- **Train batch size:** 128
- **Max prompt length:** 1024 tokens

- **Max response length:** 3072 tokens
- **Generation per question:** 8 responses (rollouts)
- **Sampling strategy:** Temperature=1
- **Optimizer:** AdamW
- **Learning rate:** $5 \times 10^{-6}$
- **Epochs:** 1
- **Reward:** RLVR

### B.1.3   TTRL

We train the TTRL for $\approx 13$ hours with the following settings:

- **Batch size:** 128
- **Micro batch size:** 4
- **Rollout batch size:** 64
- **Micro rollout batch size:** 4
- **Temperature:** 1.0
- **Total global steps:** 50
- **Generated sequence length:** 3072
- **Prompt max length:** 1024
- **N votes per prompt:** 64
- **Optimizer:** DeepSpeedCPUAdam
- **Actor learning rate:** $5 \times 10^{-7}$
- **Critic learning rate:** $9 \times 10^{-6}$
- **KL coefficient:** 0.00 (with KL loss)
- **Advantage estimator:** group_norm
- **Discount factor $\gamma$:** 1.0
- **GAE parameter $\lambda$:** 1.0
- **Nodes/GPUs:** 1 node with 4 GPUs for actor, critic, ref (A100); vLLM with 4 engines, tensor parallel size 1
- **Precision:** bfloat16
- **Reward:** Majority Voting

All hyperparameters were kept consistent across experiments, unless otherwise specified.

### B.1.4   SPURIOUS REWARD

We train the Spurious Reward with the following settings:

- **Batch size:** 128
- **Micro batch size:** 4
- **Rollout batch size:** 64
- **Micro rollout batch size:** 4
- **Temperature:** 1.0
- **Total global steps:** 50
- **Generated sequence length:** 3072
- **Prompt max length:** 1024
- **Optimizer:** DeepSpeedCPUAdam

- **Actor learning rate:** $5 \times 10^{-7}$
- **Critic learning rate:** $9 \times 10^{-6}$
- **KL coefficient:** 0.00 (with KL loss)
- **Advantage estimator:** group_norm
- **Discount factor $\gamma$:** 1.0
- **GAE parameter $\lambda$:** 1.0
- **Nodes/GPUs:** 1 node with 4 GPUs for actor, critic, ref (A100); vLLM with 4 engines, tensor parallel size 1
- **Precision:** bfloat16
- **Reward:** Spurious Reward

All hyperparameters were kept consistent across experiments, unless otherwise specified.

## B.2 EVALUATION CONFIGURATION

We evaluated the performance of all models using the official Qwen mathematical evaluation scripts. Except for the re-implemented models explicitly noted, all other models were tested directly with pre-trained weights from HuggingFace, with evaluation parameters kept consistent across all models.

- **Generation per question:** 4
- **Temperature:** 0.6, unless otherwise specified.
- **Top-p sampling:** 0.95
- **Generated max new sequence length:** 3072, unless otherwise specified.
- **Prompt type:** Math CoT — Please reason step by step, and enclose your final answer within \boxed{}.

