# OpenReview forum: "Confidence Is All You Need: Few-Shot RL Fine-Tuning of Language Models"
_ICLR.cc/2026/Conference — ICLR 2026 Conference Withdrawn Submission_

### Official Review · Reviewer_83Ki · 2025-10-20

**Soundness:** 2
**Presentation:** 3
**Contribution:** 2
**Rating:** 4
**Confidence:** 5

**Summary:**

This paper proposes a new label-free algorithm for post-training of language models. The algorithm is generally increasing the model's confidence on its output, say $\mathbb E_{y\sim p}p(y)=\sum_{y\sim p}p^2(y)$, which is easy to implement following policy gradient theorem, and can be smoothed with a hyperparameter on the weight. The experiments are conducted on extensive datasets and various models, using the metric pass@1. The results of self-confidence approach exhibit observable advantages over other label-free algorithms. The authors also provide some analysis on the experimental result.

**Strengths:**

- Clear paper writing.
- Extensive experiments.
- The RLSC loss itself is novel to the reviewer.

**Weaknesses:**

*major*:
- The main claim, "This work demonstrates that high-quality post-training can emerge not from external labels, but
from a model’s internal signal - when that signal is derived with care.", is not novel, which has been revealed in many recent papers [1,2]. This claim is especially well-known for Qwen series models, where RLSC exactly exhibits the most prominent improvement; the improvement on llama is marginal, compared with that of Qwen, as shown in figure 3.
- The main metric, pass@1, is not a strong signal to reflect the model strength. It is intuitive that increasing self-confidence can sharpen the distribution, which would clearly benefit the pass@1 metric. However, as many papers like [3] point that, the performance on pass@k might not be improved. It would be better to cover results on pass@8 or pass@16 to see whether the advantage still exists.

*minor*:
- In Section 2.2, a coefficient $2$ on the gradient is missed.

[1] Wang et al. Reinforcement Learning for Reasoning in Large Language Models with One Training Example. https://arxiv.org/abs/2504.20571

[2] Shao et al. Spurious Rewards: Rethinking Training Signals in RLVR. https://arxiv.org/abs/2506.10947

[3] Yue et al. Does Reinforcement Learning Really Incentivize Reasoning Capacity in LLMs Beyond the Base Model? https://arxiv.org/abs/2504.13837

**Questions:**

- It is not clear to the reviewer that whether there exists a clear separation between adjusting temperature and conducting self-confidence training. By decreasing temperature, the distribution of the output can also be sharpened, which aligns with the intuition of RLSC. Could the authors conduct some comparison experiments on this point?
- As shown in figure 5, the entropy would decrease as the confidence increase. So what's the difference between increasing confidence and directly decreasing entropy? Would directly adding an entropy term in the training objective achieve the same effect as RLSC, or even better?
- Section 4.2 of [1] shows that increasing entropy can enhance the reasoning power of language models, which seems a conflict to this paper's claim. Could the authors provide any clarification?

---

> ### Author Response · Authors · 2025-11-25
>
> Thank you for your thoughtful and constructive review comments. We now address each of your core concerns and questions as follows:
>
> ----
>
> ### **Weakness 1**
>
> Thank you for raising this important point regarding novelty and for the observation on model performance. We agree that the general idea of using internal signals for self-improvement is an active area of research. However, our work provides a distinct and novel contribution in two key areas:
>
> Methodological and Theoretical Novelty: While prior work like [1,2] empirically demonstrates that internal signals can be useful, they often rely on heuristic scoring functions. Our primary contribution is the derivation of a continuous, theoretically-grounded objective function (RLSC) that formally explains and generalizes the reinforcement process initiated by methods like TTRL. This provides a new mathematical framework for understanding how and why such self-reinforcement works, moving beyond empirical observation to theoretical explanation.
>
> Empirical Advancement and Generalizability: Our method demonstrates stronger and more consistent empirical results. As shown in our Table 2, RLSC achieves a 51.1% average improvement versus 29.9% for the Spurious Reward method [2] on the same benchmarks. Furthermore, we address a key limitation noted in [2]—their reported inability to achieve improvements on LLaMA and Olmo2-7B models. Our successful improvement on these models underscores our method's broader applicability and robustness.
>
> Regarding the difference between Qwen and LLaMA in Table 3, the reviewer's observation is accurate and aligns with our theoretical premise. Models like Qwen, which exhibit lower initial confidence (as seen in our Tables 1 & 2), LLaMA has a high initial confidence. Therefore, our approach of explicitly enhancing model confidence yields more significant results on Qwen. The improvement on LLaMA, while more modest, is still significant and demonstrates that the benefit of our method generalizes across model architectures, rather than being limited to a single family.
>
> In summary, our novelty lies in providing a unified theoretical foundation for self-reinforcement that is both more effective and more generally applicable than prior heuristic approaches.
>
> **Table 1. Qwen2.5-Math-7B — Top-10 Logits**
>
> | **logits** | **token id** | **token** |
> |-----------:|-------------:|-----------|
> | 12.4068 | 54 | `W` |
> | 12.3852 | 1249 | `To` |
> | 10.7775 | 5338 | `First` |
> | 10.3946 | 75 | `l` |
> | 9.8537 | 785 | `The` |
> | 9.8399 | 10061 | `Let` |
> | 9.2098 | 59 | `\\` |
> | 9.0902 | 220 | ` ` |
> | 9.0674 | 1654 | `We` |
> | 9.0657 | 12549 | `Since` |
>
>
> **Table 2. LLaMA-3.1-8B-Instruct — Top-10 Logits**
>
> | **logits** | **token id** | **token** |
> |-----------:|-------------:|-----------|
> | 30.6256 | 1271 | `To` |
> | -∞ | 4 | `%` |
> | -∞ | 8 | `)` |
> | -∞ | 9 | `*` |
> | -∞ | 7 | `(` |
> | -∞ | 3 | `$` |
> | -∞ | 1 | `"` |
> | -∞ | 5 | `&` |
> | -∞ | 0 | `!` |
> | -∞ | 2 | `#` |
> ----
>
> ### **Weakness 2**
>
> Thank you for this insightful and constructive feedback. We agree that evaluating beyond pass@1 is essential for a comprehensive understanding of a model's capabilities, as a method that merely sharpens the distribution could improve pass@1 at the expense of diversity and performance in a pass@k setting.
>
> Following the reviewer's suggestion, we have evaluated our method on pass@16 and pass@32 metrics. As now shown in Table 3 of our updated manuscript, the performance advantage of our RLSC method is maintained at these higher k values. This indicates that our approach does not simply trade diversity for sharpness but confers a genuine and robust improvement in the model's overall problem-solving ability.
>
> We have also added a discussion of this point and cited paper [3] in the related work section to properly situate our contribution in the context of this important methodological consideration.
>
> **Table 3. Pass@k Comparison across AMC23, MATH-500, Minerva, and Olympiad-Bench**
>
> | **k** | **Model** | **AMC23 (%)** | **MATH-500 (%)** | **Minerva (%)** | **Olympiad-Bench (%)** |
> |------|-----------|----------------|-------------------|------------------|--------------------------|
> | 16   | Qwen2.5-Math-7B      | 82.5           | 88.0              | 41.91            | 52.15                    |
> | 16   | Qwen2.5-Math-7B (RLSC)      | **90.0**       | **92.4**          | **56.99**        | **66.27**                |
> | 32   | Qwen2.5-Math-7B      | 90.0           | 92.6              | 52.57            | 58.24                    |
> | 32   | Qwen2.5-Math-7B (RLSC)      | **90.0**       | **93.2**          | **59.93**        | **69.84**                |
> ----
>
> (1/3 To be continued)

---

> ### Author Response · Authors · 2025-11-25
>
> ### **Weakness 3**
>
> Thank you for your careful attention. This was indeed a typo error; the correct formula is as follows.
>
> $$
> \\begin{aligned}
> \\nabla_\\theta F(p_\\theta)
> &= \\sum_y 2 p_\\theta(y | x) \\cdot \\nabla_\\theta p_\\theta(y | x) = 2 \\mathbb{E}_{y \\sim p(\\theta)} [ \\nabla _\\theta p _\\theta(y | x) )] = 2 \\mathbb{E} _{y \\sim p(\\theta)} [ \\operatorname{sg}(p _\\theta(y | x)) \\cdot \\nabla _\\theta  logp _\\theta(y | x)]
> \\end{aligned}
> $$
>
> where $\operatorname{sg}(\cdot)$ denotes the stop-gradient operator.
>
> ### **Question 1. It is not clear to the reviewer that whether there exists a clear separation between adjusting temperature and conducting self-confidence training. By decreasing temperature, the distribution of the output can also be sharpened, which aligns with the intuition of RLSC. Could the authors conduct some comparison experiments on this point?**
>
> Thank you for this insightful question, which gets to the heart of our method's contribution. We agree that lowering the sampling temperature also sharpens the output distribution, and this is a valuable baseline for comparison.
> However, as the data in our Table 4 demonstrates, the performance gains from RLSC training are significantly more pronounced than those achieved by merely adjusting the temperature. This result indicates that RLSC's benefits cannot be fully explained by simple distribution sharpening.
>
> The critical distinction is methodological:
>
> Temperature Scaling is a post-hoc, static adjustment applied during inference. It re-weights the existing distribution without updating the model's fundamental knowledge or reasoning process.
> RLSC Training is a learning process that actively fine-tunes the model's parameters. It builds upon the model's own internal reinforcement of confidence in its reasoning paths, thereby forming deeper, more reliable, and more confident internal representations.
>
> In short, while temperature adjustment changes how we sample from the model, RLSC changes the model itself. The superior performance of RLSC suggests it induces a more fundamental and beneficial alignment of the model's confidence with its competence.
>
> **Table 4. Qwen2.5-Math-7B performance with top-p = 0.95 at different temperatures**
>
> | Temperature | AIME @ 4 | Math500 @ 4 | AMC @ 4 | MMLU_Stem @ 4 | Minerva_Math @ 4 | OlympiadBench @ 4 | **Avg** |
> |-------------|-----------|--------------|----------|----------------|--------------------|----------------------|---------|
> | **0**   | 17.50 | 51.24 | 43.75 | 52.23 | 12.77 | 16.81 | **32.38** |
> | **0.1** | 7.50  | 53.20 | 43.12 | 52.12 | 11.20 | 17.66 | **30.80** |
> | **0.2** | 12.50 | 51.34 | 43.13 | 52.51 | 11.67 | 16.40 | **31.26** |
> | **0.4** | 10.83 | 50.70 | 39.38 | 50.69 | 10.47 | 15.37 | **29.57** |
> | **0.6** | 10.00 | 46.10 | 36.25 | 47.69 | 12.31 | 14.11 | **27.74** |
> | **0.8** | 11.66 | 40.75 | 26.25 | 42.80 | 9.28  | 12.66 | **23.90** |
> | **1.0** | 4.16  | 30.85 | 22.50 | 35.74 | 7.16  | 10.07 | **18.41** |
>
> ----
>
> (2/3 To be continued)

---

> ### Author Response · Authors · 2025-11-25
>
> ### **Question 2. As shown in figure 5, the entropy would decrease as the confidence increase. So what's the difference between increasing confidence and directly decreasing entropy? Would directly adding an entropy term in the training objective achieve the same effect as RLSC, or even better?**
>
> Although RLSC does achieve entropy reduction, its essence differs fundamentally from directly adding an entropy penalty. RLSC calculates gradients for whole trajectories, therefore increasing probability of most probable answers.
> By contrast, entropy regularisation indiscriminately sharpens the entire distribution, failing to distinguish between high-confidence and low-confidence inference paths.
>
> Consequently, explicitly reducing entropy values cannot leverage the cross-sample consistency signals upon which RLSC relies; the objectives of the two approaches are neither equivalent nor interchangeable in effect.
>
> We incorporated both maximising and minimising entropy into the objective function of the RLSC, with the results presented in Tables 5 and 6, overall, RLSC remains the optimal choice.
>
> **Table 5. Entropy Maximization (L = RLSC - H)**
> | Benchmark       | AIME24 | AMC23 | Math500 | Minerva_Math | MMLU_Stem | OlympiadBench | **Avg** |
> |-----------------|--------|--------|---------|----------|-----------|----------------|--------|
> | Score (%)       | 13.33  | 66.25  | 78.20   | 41.63        | 62.98     | 39.25         | **50.27** |
> | RLSC (%)       | 17.5  | 67.5  | 79.0   | 40.2        | 63.1     | 39.4         | **51.1** |
>
> **Table 6. Entropy Minimization (L = RLSC + H)**
> | Benchmark       | AIME24 | AMC23 | Math500 | Minerva_Math | MMLU_Stem | OlympiadBench | **Avg** |
> |-----------------|--------|--------|---------|----------|-----------|----------------|--------|
> | Score (%)       | 14.16  | 58.75  | 78.35   | 39.43        | 62.93     | 39.29         | **48.15** |
> | RLSC (%)       | 17.5  | 67.5  | 79.0   | 40.2       | 63.1     | 39.4         | **51.1** |
>
> ---
>
> ### **Question 3. Section 4.2 of [1] shows that increasing entropy can enhance the reasoning power of language models, which seems a conflict to this paper's claim. Could the authors provide any clarification?**
>
> Thank you for your question. While the use of these methods may seem contradictory, the role of entropy increase in that case differs from our perspective.
>
> The use of negative entropy loss during training aims to encourage exploratory behavior. [1] The study employed fixed input data; without incentivizing model exploration, the model's responses would exhibit either entirely incorrect or entirely correct outcomes to some extent. In such scenarios, the GRPO-based advantage would be zero, potentially leading to gradient vanishing during model training. Therefore, negative entropy loss is crucial for preventing such phenomena.
>
> In contrast, RLSC aims to enhance the confidence of the model's reasoning path. The training process achieves entropy reduction rather than collapse by averaging responses across multiple samples, resulting in more confident and stable model outputs.
>
> In summary, both entropy minimization and entropy maximization are effective methods for improving a model's final accuracy, but they play distinctly different roles in different training objectives.
>
> ----
>
> We extend our gratitude once more for your feedback. Acting upon your suggestions, we conducted several experiments which have significantly enhanced both the conceptual framework and empirical presentation of the paper.

---

### Official Review · Reviewer_pBtQ · 2025-10-31

**Soundness:** 3
**Presentation:** 2
**Contribution:** 2
**Rating:** 4
**Confidence:** 4

**Summary:**

This paper proposes a method for fine-tuning LLMs by using the most likely output for a given prompt. The main claim is that this method is an improvement over RL fine-tuning because it does not require a ground truth reward or reward model, and can be used in a self-supervised fashion. For each prompt the method first samples a number of responses from the model under training, then gets the log probabilities of each response under the model. The loss is chosen to reinforce the responses with higher log-probability according to the model. Experiment results on math benchmarks show that the proposed method can improve model performance, on par with RL fine-tuning and other baselines from prior works.

**Strengths:**

1. The proposed method relies of model's own log-probabilities to improve performance, removing the need for a reward as in RL fine-tuning. This simplification is a valuable contribution in cases where rewards are difficult to obtain.

2. Experiments studied a wide array of base model classes (Qwen, llama, gemma, etc.) and several common math benchmarks. The results show that the proposed method can match RL fine-tuning in model improvement.

**Weaknesses:**

1. One key issue with the proposed method of reinforcing the highest probability response is that it cannot correct cases where the most-likely response is initially wrong. As mentioned in Sec. 2.1 the motivation behind the proposed method is biasing the probability distribution  towards the most-likely response, which cannot make the necessary correction.

Such cases is exactly why external information from a reward signal is needed to improve a model.

Is this method to be used as a final fine-tuning stage after RL fine-tuning (e.g., similar to applying majority voting to an RL fine-tuned model (Sec. 2.1)? If this is the intended usage, then it would still require a reward for the RL fine-tuning.

It is interesting that by simply reinforcing the most likely response, model performance can be improved to match RL fine-tuning. However, this is more of a negative result for RL fine-tuning (and the benchmarks) than a positive one for the proposed method.

**Questions:**

In section 2.1 the paper states that "This expression is maximized when the distribution collapses to a delta function centered on a single most probable response.", then why not make the loss function for exactly this case? Clearly, this is not a desirable outcome for fine-tuning as the model's distribution would suffer entropy collapse. What does this say about the motivation of this paper?

Why is the TTRL method a natural starting point for this paper? Is it the best reward-free fine-tuning method?

In Sec. 3.1, step 2, what does "to ensure the original distribution remains unchanged" mean?

What does pass@k look like after fine-tuning using the proposed method? Is pass@1 approaching pass@k or is does pass@k collapse?

There are editing errors in the paper (for example duplicate text in Sec 3.2). Please proofread carefully.

---

> ### Author Response · Authors · 2025-11-25
>
> We are most grateful for your careful review and assessment of our work. We now address each of the questions as follows:
>
> ----
>
> ### **Weakness**
>
> We thank the reviewer for this important point. We agree that our method, by design, cannot directly replace the model's initial highest-probability response in cases where it is incorrect. However, the method operates under a key statistical assumption: across many samples, the most probable response is correct more often than not. Instances where the answer is wrong can thus be viewed as a form of "label noise", and the model is expected to correct them by learning from the majority of correct answers.
>
> As demonstrated in prior work like the TTRL paper, a model trained on majority vote labels can outperform the application of simple majority voting directly at inference time. This indicates that the learning process can indeed generalize beyond the noise and effectively learn to correct some of the errors.
>
> Our empirical results strongly support this. We observe consistent improvement across the board, suggesting that while a perfect, explicit reward signal would be ideal, our self-reinforcement approach provides a highly effective and useful signal in practice.
>
> For example, as shown in Table 1 (specifically, answer indices 3 and 7), we conducted 8 sampling runs on the same problem. The results demonstrate that, while the model has significantly increased overall confidence in its answers, there is now a difference in confidence between true and false answers which were indistinguishable for the initial model.
>
> **Table 1. Model Confidence Before vs After Training**
>
> | Index of answers | logprob_old (Before) | logprob_new (After) | Correct? |
> |------:|--------------------:|------------------:|:--------:|
> | 0     | -43.5173            | -7.0625           | Yes      |
> | 1     | -41.2783            | -7.1875           | Yes      |
> | 2     | -45.3881            | -8.8125           | Yes      |
> | **3**     | **-41.2288**            | **-12.8125**          | **No**       |
> | 4     | -51.6804            | -8.4375           | Yes      |
> | 5     | -40.9849            | -5.0000           | Yes      |
> | 6     | -49.9335            | -8.5625           | Yes      |
> | **7**     | **-52.5198**            | **-12.6875**          | **No**       |
> ---
>
> ### **Question 1. In section 2.1 the paper states that "This expression is maximized when the distribution collapses to a delta function centered on a single most probable response.", then why not make the loss function for exactly this case? Clearly, this is not a desirable outcome for fine-tuning as the model's distribution would suffer entropy collapse. What does this say about the motivation of this paper?**
>
> Thank you for pointing this out. Our general motivation is not to bring the distribution closer to collapse, but to reproduce and explain the training process or TTRL method in a more straightforward and continuous way. The loss function is derived as an interpretation of the majority voting process. One of the effects of training with labels from majority voting is a concentration of probability mass on distribution modes, which, in the limit case, can turn the distribution into a delta function. The original statement was intended to demonstrate the similarity between the TTRL method and our objective and show coherence with the idea of increasing the model’s confidence; we now see that it might be misleading and will remove it from the text. In practice, the training averages over multiple samples, so it makes the distribution sharper without collapse.
>
> The idea of applying a loss that brings the model closer to entropy collapse indeed seems counterintuitive, but this apparent contradiction is precisely the motivation of our study. TTRL research provides positive results from such training, and our goal is to provide a better theoretical explanation of this phenomenon.
>
> ----
> ### **Question 2. Why is the TTRL method a natural starting point for this paper? Is it the best reward-free fine-tuning method?**
>
> We will be honest, we don't say it is the best, but we really liked the idea and it was very convincing from the experiments. Then we asked ourselves, how to write a mathematical formulation for this method in a more detailed way, and that is how RLSC was born. Then we did the experiments and confirmed that the new method works.
>
> ----
>
> ### **Question 3. In Sec. 3.1, step 2, what does "to ensure the original distribution remains unchanged" mean?**
>
> Thank you for this question. This phrase refers to our use of the model's original predicted distribution (temperature=1) for computing log-likelihoods, rather than the temperature-adjusted distribution used for sampling. This ensures that our confidence estimates are derived from a faithful representation of the model's knowledge, free from the distortion introduced by the sampling temperature.
>
> (1/2 To be continued)

---

> ### Author Response · Authors · 2025-11-25
>
> ### **Question 4. What does pass@k look like after fine-tuning using the proposed method? Is pass@1 approaching pass@k or is does pass@k collapse?**
>
> As shown in Table 2, by comparing Pass@16 and Pass@32, we observe that our RLSC algorithm maintains its advantage. Moreover, no pass@k collapse occurs.
>
> **Table 2. Pass@k Comparison across AMC23, MATH-500, Minerva, and Olympiad-Bench**
>
> | **k** | **Model** | **AMC23 (%)** | **MATH-500 (%)** | **Minerva (%)** | **Olympiad-Bench (%)** |
> |------|-----------|----------------|-------------------|------------------|--------------------------|
> | 16   | Qwen2.5-Math-7B      | 82.5           | 88.0              | 41.91            | 52.15                    |
> | 16   | Qwen2.5-Math-7B (w RLSC)      | **90.0**       | **92.4**          | **56.99**        | **66.27**                |
> | 32   | Qwen2.5-Math-7B      | 90.0           | 92.6              | 52.57            | 58.24                    |
> | 32   | Qwen2.5-Math-7B (w RLSC)      | **90.0**       | **93.2**          | **59.93**        | **69.84**                |
>
> ----
>
> ### **Question 5. There are editing errors in the paper (for example duplicate text in Sec 3.2). Please proofread carefully.**
>
> Thank you for your careful attention, this is a typo, we will remove it.
>
> ----
>
> We greatly appreciate your thoughtful review, which has encouraged us to make our paper more accurate. Following your feedback, we conducted additional experiments and hope that their results provide clarification.

---

### Official Review · Reviewer_XpMs · 2025-11-02

**Soundness:** 3
**Presentation:** 3
**Contribution:** 3
**Rating:** 4
**Confidence:** 4

**Summary:**

This paper introduces Reinforcement Learning via Self-Confidence (RLSC), a fine-tuning method for language models that uses model prediction confidence as reward signals. Unlike conventional RL approaches that require human annotations, external reward models, or manually crafted reward functions, RLSC is self-supervised, leveraging only the model's internal probability distributions.
RLSC is efficient, requiring only 1-8 samples per problem and converging in 15-30 training steps. The method achieves significant performance improvements across mathematical reasoning benchmarks and demonstrates effectiveness across various model architectures.

**Strengths:**

- The motivation behind the proposed method is sound.
- The paper is well-organized and easy-to-follow.
- Expermental results show that the proposed method is effective.

**Weaknesses:**

- Citation format employed in the paper requires revision. For example, "Models such as DeepSeek-R1 Guo et al. (2025)" should be "Models such as DeepSeek-R1 (Guo et al., 2025)".
- Notations in Table 3 seem unclear. LLaMA-8B should be written as LLaMA-3.1-8B; Qwen-Math-1.5B should be written as Qwen2.5-Math-1.5B; Gemma-4B-pt should be written as Gemma-2-4B-pt. And the similar issue occurs from Line 081 to Line 087.
- Missing references:
[1] Wang, Yiping et al. “Reinforcement Learning for Reasoning in Large Language Models with One Training Example.” ArXiv abs/2504.20571 (2025).

**Questions:**

- There seems to be inconsistency in model selection across different backbones. For the Gemma-series model, you used the pre-trained checkpoint. And for OLMo-2-7B, which version did you use? The pre-trained version or the instruct-finetuned version? Why use Qwen2.5-Math-7B-GRPO instead of Qwen2.5-Math-7B or Qwen2.5-Math-7B-Instruct?
- During experiments, do you use any confidence calibration mechanism since models may be overconfident on some data instances?

---

> ### Author Response · Authors · 2025-11-25
>
> Thank you for your careful reading and valuable feedback, which has significantly improved the clarity of our paper. Our point-by-point responses are presented below.
>
> ----
>
> ### **Weaknesses**
>
> We are grateful for your 3 constructive suggestions. All recommendations have been incorporated: the citation format and model names in the tables have been revised, and reference [1] is discussed in the relevant section of the work.
>
> ----
>
> ### **Question 1. There seems to be inconsistency in model selection across different backbones. For the Gemma-series model, you used the pre-trained checkpoint. And for OLMo-2-7B, which version did you use? The pre-trained version or the instruct-finetuned version? Why use Qwen2.5-Math-7B-GRPO instead of Qwen2.5-Math-7B or Qwen2.5-Math-7B-Instruct?**
>
> OLMo-2-7B is in fact OLMo-2-1124-7B, where OLMo-2-1124-7B is the pre-trained version.
> As for Qwen2.5-Math-7B-GRPO, as mentioned, we fine-tuned an RL model for one epoch on Deepscaler using the training script provided by Verl [2]. We compared the performance of RLSC and GRPO. The results indicate that our method achieves comparable performance to the fine-tuning approach of GRPO.
>
> We also provide comparisons with Qwen2.5-Math-7B and Qwen2.5-Math-7B-Instruct in Table 2 of the article.
>
> ----
> ### **Question 2. During experiments, do you use any confidence calibration mechanism since models may be overconfident on some data instances?**
>
> Thank you for this question. We did not apply any confidence calibration mechanisms during our experiments, as our goal is to show that increasing only confidence can improve the overall performance. Moreover, the key advantage of RLSC is that it does not require truth labels, unlike supervised calibration methods. Incorporating unsupervised confidence calibration is a promising direction to further enhance our method's performance, and we agree it is a valuable avenue for future work.
>
> ----
>
> [1] Wang, Yiping et al. “Reinforcement Learning for Reasoning in Large Language Models with One Training Example.” ArXiv abs/2504.20571 (2025)
>
> [2] Sheng G, Zhang C, Ye Z, et al. Hybridflow: A flexible and efficient rlhf framework[C]//Proceedings of the Twentieth European Conference on Computer Systems. 2025: 1279-1297.
>
> ----
> We sincerely thank you for your careful and constructive comments, which have helped us make the paper more accurate. We hope our response addresses your concerns and clarifies the points raised to improve the score.

---

### Official Review · Reviewer_GTes · 2025-11-02

**Soundness:** 3
**Presentation:** 2
**Contribution:** 3
**Rating:** 4
**Confidence:** 2

**Summary:**

The paper proposes RLSC (Reinforcement Learning via Self-Confidence), which is a post-training method that treats a model’s output confidence as the reward, avoiding human labels, preference models, or handcrafted/verifiable rewards.
The key idea is to formalize mode sharpening, leading to a differentiable self-confidence objective and simple sequence-level losses that can be optimized efficiently. Experiments report the effectiveness of RLSC across multiple backbones on mathematical reasoning benchmarks.

**Strengths:**

- This paper converts self-confidence into a direct, differentiable objective, and training requires no external rewards or labels and uses few samples per question.
- This paper reports convergence within 15–30 steps and strong efficiency compared to TTRL’s multi-sample majority voting.
- Empirical results demonstrate that RLSC improves performance across multiple backbones and benchmarks.

**Weaknesses:**

- The method explicitly sharpens the output distribution, which could harm exploration/diversity or BoN performance. Pass@k metrics should be reported.
- Benchmarks are predominantly mathematical reasoning. Generalization to other domains (code and instruction following) is not demonstrated, limiting external validity.

**Questions:**

- The method sharpens the output distribution, which could harm exploration/diversity or BoN performance. Could you provide results on pass@k metrics to evaluate this aspect?
- The experiments focus on mathematical reasoning benchmarks. Could you provide results on other domains such as code generation or instruction following to validate the generalization of RLSC?

---

> ### Author Response · Authors · 2025-11-25
>
> We sincerely thank you for pointing out the questions regarding diversity and the effectiveness of RLSC in other domains. Our responses are as follows:
>
> ### **Weakness 1 & Question 1: The method sharpens the output distribution, which could harm exploration/diversity or BoN performance. Could you provide results on pass@k metrics to evaluate this aspect?**
>
> Thank you for your valuable question. Pass@k is indeed a very useful metric to show diversity, so we conducted a comprehensive evaluation of Pass@k according to your suggestion. The results indicate no significant impairment to diversity as shown in Table 1.
>
> **Table 1. Pass@k Comparison across AMC23, MATH-500, Minerva, and Olympiad-Bench**
>
> | **k** | **Model** | **AMC23 (%)** | **MATH-500 (%)** | **Minerva (%)** | **Olympiad-Bench (%)** |
> |------|-----------|----------------|-------------------|------------------|--------------------------|
> | 16   | Qwen2.5-Math-7B      | 82.5           | 88.0              | 41.91            | 52.15                    |
> | 16   | Qwen2.5-Math-7B (w RLSC)      | **90.0**       | **92.4**          | **56.99**        | **66.27**                |
> | 32   | Qwen2.5-Math-7B      | 90.0           | 92.6              | 52.57            | 58.24                    |
> | 32   | Qwen2.5-Math-7B (w RLSC)      | **90.0**       | **93.2**          | **59.93**        | **69.84**                |
> ----
> ### **Weakness 2 & Question 2: The experiments focus on mathematical reasoning benchmarks. Could you provide results on other domains such as code generation or instruction following to validate the generalization of RLSC?**
>
> We trained our model based on DeepSeek-R1-Distill-Qwen-7B on the DeepCoder-Preview-Dataset, employing RLSC as the objective loss function. The results are presented in Table 2.
>
> **Table 2. Performance comparison across benchmarks**
>
> | **Model** | **Training Data** | **HumanEval+** | **LiveCodeBench** | **Codeforces** |
> |-----------|-----------------|--------------------|-----------------|-----------------|
> | Baseline | * | **80.8%**           | 30.1%              | 699.3           |
> | Ours     | Code | 78.22%          | **32.43%**             | **723.6**           |
>
> We further evaluated the model on biology, chemistry, and physics-domain and general-domain benchmarks, where improvements were still observed as shown in Table 3.
>
> **Table 3.Performance Comparison on OOD Benchmarks**
>
> | **Model**   | **Training Data** | **GPQA-Diamond (%)** | **MMLU-Pro (%)** |
> |------------|--------------------|------------------------|-------------------|
> | Qwen2.5-Math-7B   | *         | 22.72                  | 25.14             |
> | Qwen2.5-Math-7B (w RLSC)       | Math          | **29.29**              | **26.58**         |
>
> ----
> We greatly appreciate your thoughtful review, which encouraged us to make our paper more accurate and provide answers on some open questions. Following your feedback, we conducted additional experiments and hope that their results will be enough to improve the score of our submission.

---

### Author Response · Authors · 2025-12-02
**Summary of key reviewer concerns for AC**

We have categorized the reviewers' key concerns into 7 areas. Through additional experiments, detailed analysis, and clarifications in our rebuttal, we believe we have thoroughly addressed these concerns and fully resolved the reviewers' issues.

----

**1. Pass@k Evaluation**

Multiple reviewers expressed concern about the lack of Pass@k evaluation results. Our supplementary experiments demonstrate that RLSC consistently outperforms the baseline model on both Pass@16 and Pass@32 metrics, proving that the confidence-boosting mechanism enhances reasoning quality without significantly compromising the model's ability to generate diverse correct solutions.

**2. General-Domain Performance**

We tested RLSC on code generation, biology, chemistry, physics, and general QA tasks. RLSC consistently outperforms baselines across all domains, demonstrating broad applicability beyond mathematical reasoning.

**3. Differences Between RLSC and Temperature Adjustment**

We compared RLSC with optimally-tuned temperature settings. RLSC consistently outperforms temperature baselines because it trains the model, whereas temperature adjustment only modifies sampling behavior without changing the model's internal distribution.

**4. Differences Between Qwen2.5-Math-7B and LLaMA3.1-8B-Instruct**

LLaMA3.1-8B-Instruct already exhibits higher baseline confidence with a more peaked distribution, leaving less room for RLSC improvement. Qwen2.5-Math-7B's lower baseline confidence allows greater gains from RLSC.


**5. The difference between direct entropy reduction and RLSC entropy reduction, and whether adding entropy to RLSC can achieve better results than RLSC alone?**

RLSC enhances confidence in the model's reasoning trajectories, while direct entropy regularization blindly sharpens token-level distributions without distinguishing between high-confidence and low-confidence generated samples. Experiments show that adding entropy terms to RLSC yields no performance improvement, confirming that RLSC's mechanism is superior to simple entropy manipulation.

**6. Effects of RLSC’s Entropy Reduction vs. Paper[1]’s Entropy Increase on Model Behavior**

The reviewer suggests a conflict with paper [1]'s entropy increase mechanism. However, these methods address different scenarios:

Paper [1]: Increasing entropy during single-sample training prevents gradient vanishing when rolling predictions consistently succeed or fail, and enhances the model's generalization capability after saturation.

RLSC: By enhancing confidence inference during multi-sample training and reducing entropy, it focuses the distribution on reliable answers. This reduces variance, improving the model's stability and reliability during the inference phase.
The approaches are different scenarios, not contradictory.

**7. Revisions to the Paper’s Descriptions and Formulas**

We sincerely appreciate the reviewers' constructive feedback. We will revise the unclear model naming conventions and correct the typo in the formulas.

----

We believe our rebuttal has thoroughly addressed all reviewer concerns with comprehensive experiments and detailed clarifications. Given the constraints of the current review process, we respectfully trust in the Area Chair's professional expertise to fairly evaluate the substance of our responses and the merit of our contributions. We sincerely thank the Area Chair and reviewers for their valuable time and effort in improving our work.

----

[1] Wang et al. Reinforcement Learning for Reasoning in Large Language Models with One Training Example. https://arxiv.org/abs/2504.20571

---

### Note · Authors · 2026-01-27

I have read and agree with the venue's withdrawal policy on behalf of myself and my co-authors.

---

### Meta-Review · Area_Chair_WghW · 2025-12-18

**Summary:**

Although the authors have partially addressed some of the issues raised, several major concerns remain outstanding:

First, to thoroughly investigate the impact of the proposed method on exploration, a more fine-grained evaluation using the pass@K metric—with a wider range of K values—should be included. The preliminary results provided during the rebuttal, which were limited to K = 16 and 32, are insufficient for a complete assessment.

Second, a more in-depth analysis and discussion are required to clarify exactly when and why the proposed method is effective. As noted by Reviewer pBtQ, the method is highly likely to fail if the initial, most-likely response is incorrect, which is often the case in more challenging benchmarks. Furthermore, the current discussion regarding the relationship between the proposed method and alternative strategies, such as temperature decay or entropy minimization, lacks the necessary depth and rigor to sufficiently motivate the proposed approach.

**Reviewer Concerns:**

The authors have partially addressed the concerns below:
- Limited problem domain (math reasoning). Generalization to others (coding, instruction following) is needed. [GTes]
- Notation and format issues. [XpMs, pBtQ]
- Missing reference. [XpMs]
- Inconsistent choice of base models. [XpMs]
- Is there any confidence calibration mechanism used? [XpMs]
- Is this method to be used after RL fine-tuning? [pBtQ]
- Lack of experiments to compare temperature adjustment and self-confidence training. [83Ki]
- Why not directly minimize the entropy? [83Ki]
- Why choose TTRL as the starting point? [pBtQ]

However, the following concerns are still outstanding:
- Potential harm to exploration. Pass@K should be reported. [GTes, pBtQ, 83Ki]
- Concern on the proposed method can not correct cases where the most-likely response is initially wrong. [pBtQ]
- The method is not well-motivated as it could lead to entropy collapse. [pBtQ]
- Limited novelty. [83Ki]
- Clarification on the conflicting claim that increasing entropy leads to better reasoning. [83Ki]

**Reviewer Scores:**

Given full discussion, Reviewers XpMs and GTes could consider changing their scores due to the partial address of their concerns like additional experiments on other domains and clarification on minor issues.
But the major concerns of Reviewer pBtQ and 83Ki still remain.

---

### Decision · Program_Chairs · 2026-01-26

Reject